# The Response of Endogenous ABA and Soluble Sugars of *Platycladus orientalis* to Drought and Post-Drought Rehydration

**DOI:** 10.3390/biology13030194

**Published:** 2024-03-19

**Authors:** Na Zhao, Jiahui Zhao, Shaoning Li, Bin Li, Jiankui Lv, Xin Gao, Xiaotian Xu, Shaowei Lu

**Affiliations:** 1Institute of Forestry and Pomology, Beijing Academy of Agriculture and Forestry Sciences, Beijing 100093, China; zhaona1019@126.com (N.Z.); sdzjh0520@163.com (J.Z.); lishaoning@126.com (S.L.); sxdxlibin@126.com (B.L.); ljk15136709537@163.com (J.L.); gaoxin2075@163.com (X.G.); 2Beijing Yanshan Forest Ecosystem Research Station, National Forest and Grassland Administration, Beijing 100093, China; 3College of Forestry, Shenyang Agricultural University, Shenyang 110866, China

**Keywords:** drought rhythm, rehydration, photosynthetic properties, endogenous abscisic acid, soluble sugar

## Abstract

**Simple Summary:**

The *Platycladus orientalis* is a coniferous gymnosperm tree species widely distributed in China and the fFar east of Russia. It has been widely introduced and cultivated in East and South Asia. Plant abscisic acid increases and affects stomatal behavior during soil drought, and prolonged drought-induced stomatal closure definitely affects the soluble sugar content of plants. We found significant correlations between gas exchange and abscisic acid content, as well as the soluble sugar content of *Platycladus orientalis* under different moisture conditions, as a result of increased drought stress in *Platycladus orientalis* due to ongoing global climate change. This helps us to reveal the mechanism of plant adaptation to drought–rehydration under different drought treatments.

**Abstract:**

To uncover the internal mechanisms of various drought stress intensities affecting the soluble sugar content in organs and its regulation by endogenous abscisic acid (ABA), we selected the saplings of *Platycladus orientalis*, a typical tree species in the Beijing area, as our research subject. We investigated the correlation between tree soluble sugars and endogenous ABA in the organs (comprised of leaf, branch, stem, coarse root, and fine root) under two water treatments. One water treatment was defined as T1, which stopped watering until the potted soil volumetric water content (SWC) reached the wilting coefficient and then rewatered the sapling. The other water treatment, named T2, replenished 95% of the total water loss of one potted sapling every day and irrigated the above-mentioned sapling after its SWC reached the wilt coefficients. The results revealed that (1) the photosynthetic physiological parameters of *P. orientalis* were significantly reduced (*p* < 0.05) under fast and slow drought processes. The photosynthetic physiological parameters of *P. orientalis* in the fast drought–rehydration treatment group recovered faster relative to the slow drought–rehydration treatment group. (2) The fast and slow drought treatments significantly (*p* < 0.05) increased the ABA and soluble sugar contents in all organs. The roots of the *P. orientalis* exhibited higher sensitivity in ABA and soluble sugar content to changes in soil moisture dynamics compared to other organs. (3) ABA and soluble sugar content of *P. orientalis* showed a significant positive correlation (*p* < 0.05) under fast and slow drought conditions. During the rehydration stage, the two were significantly correlated in the T2 treatment (*p* < 0.05). In summary, soil drought rhythms significantly affected the photosynthetic parameters, organ ABA, and soluble sugar content of *P. orientalis*. This study elucidates the adaptive mechanisms of *P. orientalis* plants to drought and rehydration under the above-mentioned two water drought treatments, offering theoretical insights for selecting and cultivating drought-tolerant tree species.

## 1. Introduction

Water profoundly influences the distribution of plant species, their productivity, and the structure and functioning of ecosystems [1,2]. Although some woody plants can recover from severe droughts [3], the persistent occurrence of widespread forest die-offs globally remains associated with drought occurrences [4,5]. Therefore, it is essential to clarify the physiological mechanisms associated with the impact of drought on plants and the subsequent recovery process post-drought. Leaf photosynthesis is the foundation of plant yield formation, and chloroplasts convert light energy into chemical energy, which is used to capture and assimilate carbon dioxide (CO_2_) to form carbohydrates [6]. Consequently, it is necessary to determine and quantify the extent of tolerance and recovery of plants responding to drought and post-drought rehydration.

Drought-exposed plants reflect specific characteristics that may augment plant survival and resilience to climate change [7,8]. When drought stress occurs, plants exhibit an inherent ability to adapt to this water-deficient environment through various regulatory mechanisms. The repercussions of drought stress on plants can be evaluated by monitoring physiological or biochemical indicators [9]. The enhancement of photosynthetic physiological performance facilitates the accumulation of organic matter in plants [10]. However, when a plant is subjected to drought conditions, the net photosynthesis, the water potential, and the stomatal conductance are altered, making photosynthesis a crucial indicator of plant stress resilience [11]. Upon re-watering following drought stress treatments, the decline in photosynthetic parameters, mainly influenced by stomatal limitation in plants, quickly returns to or even surpasses the pre-treatment level, resulting in a compensatory effect [12].

Endogenous plant hormones, as organic substances produced by the plant’s metabolism, are closely linked to plant growth, development, and stress tolerance [13]. Among them, abscisic acid (ABA) is considered to be one of the most crucial signal regulators in plants in response to drought stress, which can promote stomatal closure as well as the expression of stress-responsive genes [14]. When drought occurs, plants initially try to minimize excessive water loss from their leaves; hydraulic signaling leads to a decrease in transpiration rate, and plants rapidly activate root-to-stem chemical signals to partially or completely close stomata [15]. Hydraulic signaling causes stomatal closure prior to ABA to avoid excessive plant water loss and to reduce the risk of xylem embolism (bubbling and cavitation of xylem conduits). Therefore, hydraulically mediated stomatal closure is just an early plant response to drought, buffering against the risk of reduced water potential and xylem embolism. Subsequently, ABA is transported upward from the roots and accumulates in the leaves, augmenting stomatal closure. In this process, ABA assumes a critical role in the regulatory network governing stomatal behavior [16].

Non-structural carbohydrates include sugars and starch that are essential to plant metabolism, and their roles in drought stress are thought to be critically important [16,17]. In this context, during the early stages of drought, NSC concentrations might increase as growth decreases, preceding the decline in the photosynthetic rate [18]. However, as drought persists, the decline in photosynthesis exceeds respiration, leading to a decline in NSCs due to an imbalance in carbohydrate production and utilization [19]. In addition, different organs and tissues of trees carry out different functions and have different stores of carbohydrates [20,21]. Generally, the supply of photosynthetic products from the top to the bottom of the plant conforms to the proximity principle, prioritizing allocation to the most essential organs [22]. During mild drought, plants prefer to store NSCs in their roots [23,24], whereas severe drought conditions prompt a higher accumulation of NSCs in aboveground tissues [25]. Higher trees have long carbon source-to-storage distances. When exposed to drought events, these trees will increase the viscosity of phloem sap, which may lead to impaired carbon assimilation transport channels from leaves to roots and aggravate the uneven distribution of NSC concentrations across tree organs [26,27,28]. Previous studies have mainly focused on changes in a single indicator of ABA or osmoregulatory substance content in a tree species subjected to drought and rehydration [29,30]. However, there remains a gap in the understanding of the crosstalk mechanisms between ABA and mobile carbon sources in plant organs responding to drought stress and subsequent rehydration recovery treatments. Therefore, understanding the response of ABA and soluble sugars to drought stress in various tree species can thus aid in exploring the effects of drought on plant growth alongside the mechanisms of adaptation of physiological processes to drought.

Currently, studies have mostly focused on the characteristics of plant photosynthetic traits or alterations in ABA content under drought stress [31,32]. However, there remains a lack of clarity regarding the changes in ABA content across various plant organs and their impact on stomatal behavior and photosynthetic carbon uptake under dynamically changing water conditions. Higher, intense, but shorter duration drought stress under natural conditions significantly inhibits the rapid development of growth and photosynthesis and causes plant water failure. However, drought events with a less intense but longer duration often occur at the beginning of the tree growth season in northern China. Such drought events will cause a continuous reduction in the speed and amount of carbon assimilation via stomata and even lead to a carbon balance imbalance and death.

In this study, we set up the rapid drought–rehydration group (T1) and the slow drought–rehydration group (T2) with soil moisture as test materials. The investigation involved meticulous measurements and analyses of plant photosynthetic parameters alongside assessments of endogenous ABA and soluble sugars within various plant organs (leaves, branches, stems, thick roots, and thin roots) under different water backgrounds. The primary objective was to address the following research questions: (a) What are the responses of ABA and soluble sugar content in the young *Platycladus orientalis* trees to different treatments of soil moisture change rates? (b) Is there a synergistic relationship between ABA and the soluble sugar content under moisture variability? (c) Does the rate of soil moisture change regulate this synergistic relationship?

## 2. Materials and Methods

### 2.1. Study Area and Material Selection

The study area was located in the Institute of Forestry and Fruit Tree Research, Beijing Academy of Agricultural and Forestry Sciences (39°59′35″ N, 116°13′13″ E), at an altitude of about 88 m. The geographical region experiences a temperate continental monsoon climate characterized by atmospheric temperatures ranging from −8 to 31 °C, an annual frost-free period of 180–200 d, and an average annual precipitation of approximately 600 mm [33].

This study utilized young three-year-old potted trees of *Platycladus orientalis* as the research object (Table 1). The pots measured 35 cm in both height and diameter at the top. The cultivation substrate utilized was locally sourced field-cultivated soil, ensuring a consistent soil capacity of around 1.21 g/cm^3^ for each pot (Table 2). In the previous year, before conducting the water treatments, the young trees were placed in a rain shelter for sapling cultivation; management and maintenance followed. Throughout the testing phase, rigorous control over other environmental factors, such as light conditions, ambient temperature, and humidity, was maintained to ensure uniformity. We cautiously selected 50 vigorous seedlings of uniform specifications from all plants for the experimental trials in June of the following year. In Beijing, in June, the beginning of the tree-growing season, the average monthly precipitation is 12 mm, which easily causes seasonal drought stress on trees. Therefore, we set up two water treatments (detailed explanations in 2.2 Experimental design) at the beginning of the growing season (May–June).

### 2.2. Experimental Design

Prior to commencing the experiment, five young sidecar potted trees with soil water content maintained at field capacity (FC) for a long period (at least two months) were selected. The water supply was halted until the top leaves of young trees started to wilt, as it is defined as the wilting node, and the volumetric soil water content (SWC) at the wilting coefficient was determined. The potted young trees were randomly divided into two experimental treatment groups, and the soil moisture in the two groups of test seedlings was maintained at FC at the beginning of the experiment. In one water treatment, named the rapid drought–rehydration treatment (T1), potted young trees were stopped from irrigation until the drought-wilting nodes of each species were reached, which was defined as a fast drought phase. Subsequently, rewatering the above tree saplings exposed to fast drought was carried out, and it was ensured that the SWC of potted trees reached the FC and was maintained for the potted trees thereafter. The other water treatment was named the slow drought–rehydration treatment (T2). In the slow drought phase of this treatment, the potted young trees with well-watered conditions suffered from the continuous deficit irrigated mode. The daily watering amount was 95% of the previous day’s water consumption based on weighing the loss of one sapling for two days at 17:00 to 18:00 every other day. This approach led to a gradual, slow drought of the potting soil. When the young trees in each pot reached the SWC, their respective wilting points (the SWC at the wilting coefficient of each sapling was the same in both T1 and T2 but differed in the arrival times), rehydration after slow drought was executed immediately until the potting SWC reached FC. The potting soil moisture was continuously maintained at the FC during the rehydration phase.

Under the two water treatments, the potting SWC of T1 and T2 for *P. orientalis* seedlings declined during the drought phase until the drought treatment was suspended when the leaf water potential at the pre-experimental wilting coefficient was reached (Figure 1). The SWCs for *P. orientalis* at the fast and slow drought phase endings were 8.4% and 8.8% for T1 and T2, respectively. The rate of soil moisture reduction was 1.4% per day for the SWC from 27.5% to 8.4% during the drought phase of T1 and 0.6% per day for the SWC from 27.2% to 8.8% during the drought phase of T2. During the post-drought rehydration recovery phase, the soil moisture SWC of young potted trees recovered and was maintained at the field water holding capacity (26.5% to 28.2%).

### 2.3. Photosynthetic Parameter Assay

Six saplings of potted *P. orientalis* at different stages of T1 and T2 were selected for each treatment, and three fully mature leaves with good growth in the upper middle part of each plant were chosen to be measured with the photosynthetic parameters, such as the net photosynthetic rate (P_n_, μmolm^−2^s^−1^), transpiration rate (T_r_, mmolm^−2^s^−1^), stomatal conductance (G_s_, molm^−2^s^−1^), and intercellular CO_2_ concentration (C_i_, μmol/mol), using a handheld CI-340 portable photosynthesizer (CID, VAN, WA, USA) from 9:00 to 11:00 a.m. on a typical sunny day. Light intensity was simulated using an external light source based on the light saturation point modeled from the pre-experiment (photosynthetically active radiation (PAR) of 1000 μmolm^−2^s^−1^).

### 2.4. Endogenous ABA Content Assay

After determining photosynthetic characteristics, three saplings of *P. orientalis* were separated according to leaves, current year twigs, stems, thick roots, and thin roots (≥2 mm in diameter for thick roots and <2 mm in diameter for thin roots). The samples were immediately wrapped in tin foil, brought back to the laboratory in liquid nitrogen, and stored in a −80 °C refrigerator. The fresh samples were rapidly pulverized using a high-speed pulverizer and stored in a refrigerator at −80 °C for the subsequent determination of ABA content. The ABA content of each organ of the plant was determined by an enzyme-linked immunosorbent assay (ELISA). A total of 0.2 g of the sample was weighed, and 1.6 mL of phosphate buffer was added to the homogenate. The mixture was then centrifuged at 8000× *g* for 10 min, and the supernatant was extracted for ABA determination [34]. The experiment was conducted with three biological replicates. The ABA content of the whole plant was calculated by considering the ABA content of each organ and the biomass proportion of each organ.

### 2.5. Soluble Sugar Content Assay

The samples were placed in a desiccator at 75 °C for 48 h and ground to a fine powder. Then, 0.1 g of the powdered sample was dissolved in 10 mL of ethanol solution (80%), shaken in a water bath at 80 °C for 10 min, and centrifuged at 3000× *g* for 10 min to obtain both the supernatant and sediment. The samples were placed in a desiccator at 75 °C for 48 h and ground to a fine powder. A total of 0.1 g of the powdered sample was then dissolved in 10 mL of ethanol solution (80%), shaken for 10 min in a water bath at 80 °C, and centrifuged at 3000× *g* for 10 min to obtain both the supernatant and sediment. The soluble sugar content was quantified from the diluted supernatant tubes using an anthrone-sulfuric acid coloring reagent (0.1% (*m*/*v*) in 98% sulfuric acid), and the absorbance was read on a spectrophotometer at the suggested wavelength of 620 nm [35].

### 2.6. Data Analysis

The data were analyzed using Microsoft Excel 2019 software (Microsoft Corporation, Washington, DC, USA) and SPSS version 25.0 software (SPSS, Inc., Chicago, IL, USA). Independent sample *t*-tests and one-way ANOVAs were utilized to analyze the differences between the two treatments, as well as between the same treatments at each sampling point. The differences in the photosynthetic characteristics and ABA content were analyzed for significance (*p* < 0.05). A Pearson’s correlation analysis was performed for photosynthetic characteristics and ABA content. Plotting was performed using Origin Pro 2021 software (Origin Lab, Northampton, MA, USA).

## 3. Results

### 3.1. Photosynthetic Performance under Different Drought–Rehydration Treatments

Drought stress significantly (*p* < 0.01) reduced the P_n_ and G_s_ of *P. orientalis*. The decrease in P_n_ of *P. orientalis* in T1 (57.30%) was significantly greater compared to T2 (37.25%), while G_s_ showed the opposite trend to P_n_ under the different drought treatments. Under T1 and T2, the Tr and Ci of *P. orientalis* showed the opposite performance. In *P. orientalis* under T2, Tr decreased by 55.56% at the S2 (completion of drought) compared with that of S1 (beginning of drought), while T1 did not show a decreasing trend. In the T1 *P. orientalis*, the Ci increased by 12.14% at the S2, and in the T2 *P. orientalis*, the Ci decreased by 45.50% at the S2. This indicates that different drought treatments resulted in differences in the photosynthetic physiology of *P. orientalis*. Specifically, the T1 *P. orientalis* was restricted by the stomata factor, while the T2 *P. orientalis* was limited by both stomata and non-stomata factors.

There were significant differences (*p* < 0.01) between the T1 and T2 *P. orientalis* in terms of P_n_, G_s_, T_r_, and C_i_. The T1 *P. orientalis* exhibited partial recovery in P_n_, G_s_, and T_r_ following drought stress relief, whereas the T2 *P. orientalis* did not recover from drought and rehydration, leading to a sustained decline in the FC. In the T1 *P. orientalis*, the C_i_ decreased to 71.95% and 91.76% of the S1 at S3 (3 days of rehydration) upon S4 (6 days of rehydration), respectively. Conversely, in the T2 *P. orientalis,* the C_i_ increased (Figure 2).

### 3.2. Effects of Different Drought Treatments on Endogenous Hormone Abscisic Acid Content

The whole plant ABA content increased with different soil moisture stress treatments (Figure 3). The ABA increment was higher in the T2 drought wilt node (25.98%) compared to T1 (8.91%) at the S1 (beginning of drought). During the rehydration stage, the T1 ABA content continued to accumulate and peaked at the S4, while the T2 maintained a higher ABA content but decreased by 4.58% at the S4 compared to the S2.

Both T1 and T2 drought stresses significantly (*p* < 0.05) increased the ABA content of leaves, twigs, and thick and thin roots. The increase in T2 was greater than that in T1, and the changes in ABA content of *P. orientalis* stems in T1 and T2 showed opposite trends. T1 fine roots exhibited the largest ABA increment of 33.83%, followed by twigs (32.36%) and coarse roots, with the smallest increase in leaf ABA content. T2 twigs had the largest ABA increment of 55.92%, followed by fine roots (47.27%), and the smallest increase was evident in leaf ABA content.

During the water recovery stage, the ABA contents of leaves, twigs, and stems of the T1 *P. orientalis* increased slightly at the S4 compared with the S2. Meanwhile, the ABA contents of roots did not change significantly (*p* > 0.05). For the T2 *P. orientalis*, the ABA content of twigs and thick roots declined, reaching 93.12% and 87.10%, respectively, of those at the S2 by the S4. However, these levels were still significantly higher than those before the drought stress. The ABA content of leaves and stems decreased and then increased. The ABA content of fine roots of T2 *P. orientalis* did not change significantly (*p* > 0.05).

### 3.3. Effects of Different Drought Treatments on the Soluble Sugar Content

The whole-plant soluble sugar content of T1 and T2 *P. orientalis* increased conspicuously (*p* < 0.05) after they were subjected to drought stress. At the S2, the increase of the whole-plant soluble sugar content of T1 (57.3%) was greater than that of T2 (40.23%). During the rewatering stage after the drought, the whole-plant soluble sugar content of the T1 and T2 *P. orientalis* continued to decline compared to that at the S2, reaching 77.9% and 78.75%, respectively, at the S2. However, these levels were still higher than those at the S1 in general.

Both T1 and T2 drought stresses significantly (*p* < 0.05) increased the ABA contents of leaves, twigs, coarse roots, and fine roots of *P. orientalis*. The soluble sugar content of fine roots in all organs of *P. orientalis* increased the most in T1, by 145.7%, followed by coarse roots and twigs, with the smallest increment observed in the leaves. In T2, the soluble sugar content of the stem of the *P. orientalis* did not change significantly at the S2 compared with that before drought, while it increased significantly in T1. Specifically, only twigs (176.43%) and fine roots (371.94%) of *P. orientalis* showed a greater increase in soluble sugar content compared to T1.

Comparing the soluble sugar content of above-ground organs during the rewatering stage, the soluble sugar content of branches and stems of the T1 and T2 *P. orientalis* at the S4 (6 days of rewatering) decreased compared with that at the S2 (completion of the drought). In T2, twigs did not recover to the pre-treatment level, while the other above-ground organs basically recovered to the pre-treatment level (ranging from 74.97 to 114.74%). In the comparison of underground organs, the fine roots of the T1 *P. orientalis* at the S4 were lower than at the S2 but recovered to the pre-treatment level. On the other hand, the T2 *P. orientalis* at the S4 showed a decrease of 41.25%, and the difference between the T1 *P. orientalis* at the S4 and the end of stem drought was essentially the same (52.23%). However, the fine roots at the S4 of the T2 *P. orientalis* did not recover to the pre-treatment level. Regarding the transportation organ stem, T1 recovered to the pre-treatment level after rehydration recovery, while the T2 stem did not change significantly (Figure 4).

### 3.4. Correlation between Soluble Sugar and Abscisic Acid Content

According to the results of the correlation between the ABA content and soluble sugar content of each organ of *P. orientalis* and each photosynthetic parameter, the T1 and T2 *P. orientalis* drought stage, rehydration stage Pn, Gs, and the whole plant ABA content and soluble sugar content were selected to make a fitting curve; the results are shown in Figure 5.

During the drought stage, there was a strong correlation (*p* < 0.05) between the photosynthetic parameters and ABA content in the T1 and T2 *P. orientalis*, both of which were significantly negatively correlated (Figure 5A,C). However, there was no significant correlation between the ABA content and T_r_ and C_i_ (*p* > 0.05). The correlation coefficients between the ABA content and photosynthetic parameters in the T2 *P. orientalis* were larger than those in the T1 *P. orientalis*. In the rehydration stage, there was no significant correlation (*p* > 0.05) between the photosynthetic parameters and ABA content in the T1 and T2 *P. orientalis* (Figure 5B,D). During the drought stage, there was a strong correlation between the photosynthetic attributes and soluble sugars in both the T1 and T2 *P. orientalis* (Figure 5E,G). However, during the rehydration stage, there was no significant correlation (*p* > 0.05) between the soluble sugar content and photosynthetic parameters in both the T1 and T2 *P. orientalis* (Figure 5F,H).

During the drought stage, there was a positive correlation (*p* < 0.05) between ABA content and soluble sugar content in both the T1 and T2 *P. orientalis*. The correlation coefficients between the ABA content and soluble sugar content in the T1 *P. orientalis* were greater than those in the T2 *P. orientalis* (Figure 5I). In the rehydration stage, only the T2 *P. orientalis* ABA content was positively correlated with soluble sugar content (*p* < 0.01) (Figure 5J).

## 4. Discussion

### 4.1. Response of Photosynthetic Characteristics to Different Drought Treatments

The limitation of plant growth by drought is primarily caused by the disruption of the plant carbon balance, which heavily relies on photosynthesis. With the decrease of soil water, the stomatal conductance of plant leaves decreases sharply in order to reduce water transpiration (Figure 2II). The limitation of photosynthesis due to water deficit is not solely attributed to stomatal closure. However, a combination of stomatal and nonstomatal factors contributes to the weakening of photosynthesis under moderate and severe drought conditions [36]. In this study, the P_n_, G_s_, and T_r_ were significantly reduced in both the T1 and T2 *P. orientalis* after drought stress, indicating that the photosynthetic characteristics of *P. orientalis* were sensitive to water deficits. The consistent trends of the G_s_, P_n_, and T_r_ basically suggested that the reduction in photosynthetic capacity was influenced by stomatal limitation. Additionally, the P_n_ decreased more in the T1 *P. orientalis* than in T2, possibly due to the ability of T1 to rapidly perceive drought and maintain a lower photosynthetic regime to minimize photosynthetic organ damage under intense and rapid drought treatments. The rapid water dissipation in T1 limited plant growth and hindered respiration [37]. Under slow drought, young trees gradually perceived soil water stress, developed the ability to adapt to gradual drought habitats, and became less sensitive to water stress. When T1 and T2 reached the same drought wilting point, the effect of T1 on the plant net photosynthetic rate was greater than that of T2. Drought stress significantly increased the C_i_ of T1 and decreased that of T2, indicating opposing trends in the inter-cellular CO_2_ concentrations of T1 and T2, suggesting differences in the limiting factors to which T1 and T2 are subjected. T2 is subjected to more stomatal limitation than T1.

Typically, plants that undergo severe water deficit recover 40–60% of their maximum photosynthetic rate the day after rewatering and continue to do so over the next few days [38]. However, the maximum photosynthetic rate is not always restored [10,39]. Despite this, the P_n_, G_s,_ and T_r_ did not fully recover after rewatering, as the damage to the photosynthetic organs during drought stress proved challenging to repair. The photosynthetic parameters of T1 recovered after rewatering, while T2 continued to decline post-rehydration. The *P. orientalis* from T1 exhibited a positive response to post-drought rewatering, actively regulating its photoprotective mechanisms during drought [40]. Conversely, T2 sustained more irreversible damage to photosynthetic organs during drought, failing to recover with rehydration. The persistent decline in stomatal conductance after rehydration in the T2 *P. orientalis* limits photosynthetic recovery but enhances intrinsic water use efficiency.

### 4.2. Response of Endogenous Hormone ABA Content to Different Drought Treatments

Abscisic acid acts as a stomatal inhibitor in plants under drought stress to diminish water loss [41]. Enzymes involved in ABA biosynthesis are expressed in the vascular tissues of roots, stems, and leaves in response to stresses such as drought [42]. The perception of drought stress at the root tip is thought to stimulate the synthesis of ABA, which is transported above ground via the transpiration stream and provides an “early warning” stress signal to the canopy [43]. In this study, the whole-plant ABA content of *P. orientalis* was augmented significantly after drought stress, which positively affected the plant drought response. This increase was conducive to the closure of leaf stomata and the reduction of the transpiration of *P. orientalis*, all aimed at improving its water retention. Additionally, the whole-plant ABA accumulation of the T2 *P. orientalis* was notably higher than that of T1. The alteration of transpiration under the rapid reduction of soil moisture may have led to the slow rate of plant xylem water conductivity, which impeded the production and transport of ABA [44]. While the slow loss of soil moisture causes plants to remain under drought stress for a long period without recovery, plants need to continuously release drought signals to synthesize ABA for transport to the leaves to maintain cellular osmotic potential to cope with slow drought [18]. The whole-plant ABA content of the T2 *P. orientalis* decreased slightly during the rewatering stage after the drought, indicating that the chemical signal regulation of the plant was weakened after the drought stress was lifted, and the synthesis of ABA subsided. However, the decrease in ABA was small, and the ABA content remained high compared to that before drought stress. Even the whole-plant ABA content of T1 continued to rise, indicating that the plant could still regulate the leaf stomata through the elevation of ABA content in the rehydration period with strong physiological activity.

In this study, the ABA content of leaves, twigs, coarse roots, and fine roots of both the T1 and T2 *P. orientalis* exhibited a marked increase following exposure to drought stress. Nevertheless, noteworthy distinctions surfaced in the degree of augmentation within each organ between T1 and T2. The elevation of ABA in the leaves, twigs, and both thick and thin roots of the *P. orientalis* under T2 significantly surpassed that observed in T1. This discrepancy can be ascribed to the prolonged stress endured by T2, indicating an extended duration of exposure to adverse conditions and, consequently, a heightened degree of chemical signal regulation [18]. The highest ABA content was found in thick roots both before and after drought treatments. This phenomenon is attributable to the slender tubular cells and the plug-edge structure of the threaded membranes characterizing, conferring a diminished efficiency in xylem transport and advocating a more conservative resource utilization strategy [45].

During the recovery phase from drought, the ABA content in the thick roots of the T2 *P. orientalis* exhibited a decreasing trend, while the ABA content in other organs had no discernible changes. This implies a partial alleviation of stress on thick roots post-rehydration, underscored by their superior recovery capacity compared to other organs. The data also revealed that following rehydration, the synthesis of ABA in the root system was curtailed, contributing substantially to the swift recovery of plants post-drought [46]. Throughout the experiment, the ABA content within the stems exhibited negligible variations in comparison to other organs, maintaining a stable profile. This suggests a lack of sensitivity in stems to alterations in water conditions.

### 4.3. Response of Soluble Sugar Content to Different Drought Treatments

Sugars stand as a fundamental carbon reserve for the growth, development, and defense mechanisms of higher plants. They represent the outcome of photosynthesis, orchestrating their journey from source to sink through the phloem [47]. Amidst drought conditions, plants trigger stress responses, culminating in a surge of soluble sugar content across every organ within the plant structure [48]. In the current investigation, the most substantial escalation in soluble sugar content was noted in the main stem and thick roots. These structural components, pivotal for water transportation and conduction, exhibit heightened sensitivity under drought compared to other organs. To counteract this stress, the plant strategically mobilizes sugars to uphold the water potential gradient in the phloem, which is in agreement with the observations made by Cernusak [49]. While leaves experienced a more moderate increase in soluble sugars under drought stress, fine roots exhibited a pronounced surge. The above results suggest that plants preferentially transport soluble sugars to roots in order to ensure sustenance of root basal metabolism, root pressure, and efficient water uptake. Soluble sugars are involved in a variety of processes to mitigate the adverse effects of drought stress [50]. The increase in soluble sugar content was higher in T1 than in T2 in the *P. orientalis*. Indeed, the response of non-structural carbohydrates (NSCs) to drought is very complex and depends greatly on the trade-off between carbon supply (i.e., photosynthesis) and carbon demand (i.e., growth, respiration, osmotic adjustment, and the maintenance of hydraulic functional traits) [51].

The soluble sugar content of the T1 and T2 *P. orientalis* decreased to varying degrees after drought rewatering. In this study, we observed the persistent functionality of dry and thick roots during the recovery phase after drought rewatering, effectively facilitating the transportation and conduction of water back to the leaves. Throughout the experimental period, noticeable disparities in the soluble sugar content were discernible across organs subjected to T1 and T2. Concurrently, the soluble sugar content in each organ of the T1 group surpassed that in the T2 group. This discrepancy implies that relative to T2, *P. orientalis* seedlings exhibited a heightened demand for soluble sugar under T1. The rapid water loss restricts seedling growth and impedes respiration, necessitating the role of soluble sugars as an osmotic regulator to maintain osmolality in the cells of the organs and support various metabolic activities. Remarkably, the soluble sugar content in the main stem increased under T1. The rapid decrease in soil moisture and transpiration led to embolisms in the xylem of the seedlings, which impeded the transmission of water and soluble sugars, resulting in the continuous accumulation of soluble sugars in the main stem [52]. In contrast, the soluble sugar content in the main stem of the T2 group remained relatively stable. The gradual loss of water did not significantly disrupt the function of xylem transport channels. Instead, it concentrated soluble sugars in the leaves to maintain cellular osmotic potential. Subsequently, these sugars were swiftly transported to the roots, where they were stored until the appropriate environmental conditions allowed for tissue and organ reconstruction, facilitating overall plant recovery. Notably, during the post-drought recovery from embolism, NSCs may constitute a major component of the osmotic force, facilitating water flow into the embolized conduit and expediting the hydraulic recovery process [53,54].

### 4.4. Correlation between Endogenous Hormone ABA and Soluble Sugar Content under Different Drought Treatments

Drought induces alterations in the content of endogenous hormones, thereby regulating stomatal behavior. It has been suggested that stomatal conductance exhibits a highly significant and negative correlation with the ABA content in leaves. Abscisic acid, in turn, governs the opening and closing of stomata under drought-stress conditions [55]. In addition to its role in stomatal regulation, ABA directly impacts the photosynthetic process by decreasing the electrical potential of chloroplast cell membranes, consequently diminishing photosynthetic electron transfer [56]. In this study, significant negative correlations were identified between the P_n_, G_s_, and the content of ABA and soluble sugars in *P. orientalis* after exposure to drought stress. The elevation of ABA resulted in a decline in the P_n_ and G_s_. This, in turn, prompted the plant to increase soluble sugar levels to maintain cellular osmotic pressure, aligning with the findings of numerous studies [57]. Under soil drought conditions, plants communicate drought information to the above-ground part of a plant through the chemical signal ABA. This actively reduces stomatal conductance, leading to a decrease in photosynthetic characteristics such as P_n_ and transpiration rate, effectively balancing plant water utilization [58]. We analyzed the correlation between the photosynthetic rate and ABA in the fast and slow drought of *P. orientalis*. Figure 5 shows that during T2, the photosynthetic rates of plants were proved to be more closely related to ABA and soluble sugar; that is, slower reductions in soil moisture can induce a significant mutual regulation between ABA and soluble sugar.

Chen et al. [59] found that sugar signaling, whether directly or indirectly, interacts with other signals, including hormones and redox-mediated processes, to regulate plant development and stress responses. However, the involvement of endogenous ABA in carbon starvation and its role in regulating plant sugar metabolism in relation to growth remains unclear. Morita-Yamamuro et al. [60] also revealed that stress induces carbon starvation in plants, which correlates with endogenous ABA levels and sensitivity in Arabidopsis. It has been demonstrated that ABA acts as an inducer of reactive oxygen species (ROS) production, leading to stomatal closure in plant leaves under drought conditions. However, ROS are not directly correlated with leaf-soluble sugar levels [61]. Asad et al. [62] found a significant positive correlation between ABA and soluble sugar in plant organs. Their findings decipher that drought stress prompts synergistic changes in sugar and ABA, providing stress mitigation benefits and promoting sugar accumulation. These results align with the outcomes of our experimental study.

In this investigation, the correlation coefficient between the soluble sugar content and ABA content in the T2 *P. orientalis* surpassed that of T1, indicating that the severity of drought stress influences the strength of the synergistic relationship between the soluble sugar content and ABA. This can be attributable to the role of ABA as a stress-reversal hormone, actively promoting the formation of soluble sugars [63,64]. The rapid and severe water imbalance may cause mechanical damage to the transport tissues, leading to metabolic disorders or the formation of xylem cavitation embolism [36]. This prevents soluble sugars from performing their osmoregulatory roles and being converted to starch for storage. The rapid accumulation of ABA in sapotaceae under prolonged slow drought was followed by a synergistic acceleration of the rate of soluble sugar conversion, which increased the water potential of sapotaceae seedlings to maintain a certain osmotic pressure [64]. In contrast, prolonged drought stress slows down the utilization of soluble sugars by seedlings, intensifying the accumulation of soluble sugars even further. This results in a more pronounced synergistic change in the content of ABA and soluble sugars under prolonged, slow drought conditions. In conclusion, different levels of drought stress influence the synergistic relationship between plant ABA and soluble sugars and lead to alterations in plant drought stress response and post-drought resilience.

## 5. Conclusions

It was found that drought has a significant effect on various physiological and biochemical indices of plants. Using *P. orientalis* as the experimental plant, the effects of drought stress on typical tree species in Beijing were determined by measuring the relevant indices such as the photosynthetic parameters, endogenous ABA concentration, and soluble sugar concentration under two different water treatments. The study revealed that different water treatments exerted varying effects on various physiological attributes of *P. orientalis*. Under drought stress, the photosynthetic characteristics of *P. orientalis* diminished while the ABA and soluble sugar content increased. A clear synergistic relationship between the ABA and soluble sugar content was evident, with the correlation being even more robust due to the mechanical damage to the xylem caused by rapid drought. After rehydration, the indices either fully or partially recovered, attributable to the damage sustained by plant organs and the regulation of the hydraulic signaling and other chemical signals. In the future, there is a need to delve into the internal structure of the plants and the synergistic regulation of multiple signals and photosynthetic properties.

## Figures and Tables

**Figure 1 biology-13-00194-f001:**
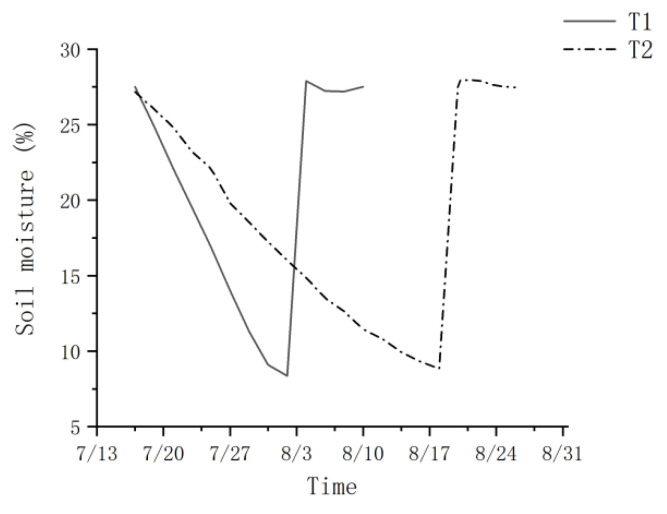
Soil water content changes under different drought treatments. T1 is the fast drought–rehydration group, and T2 is the slow drought–rehydration group.

**Figure 2 biology-13-00194-f002:**
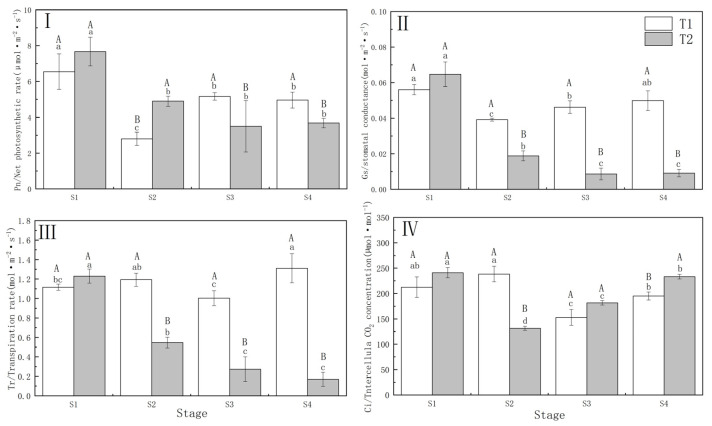
(**I**) Net photosynthetic rate (P_n_) of Platycladus orientalis during drought and rehydration periods; (**II**) Stomatal conductance (G_s_); (**III**) Transpiration rate (T_r_); (**IV**) Different letters of CO_2_ concentration (C_i_). T1: Rapid drought–rehydration group; T2: Slow drought–rehydration group. Different letters indicate significant differences in measurement indicators between the treatment group (*p* < 0.05), lowercase letters indicate differences in different stages of the same treatment, and uppercase letters indicate differences in different treatments at the same stage.

**Figure 3 biology-13-00194-f003:**
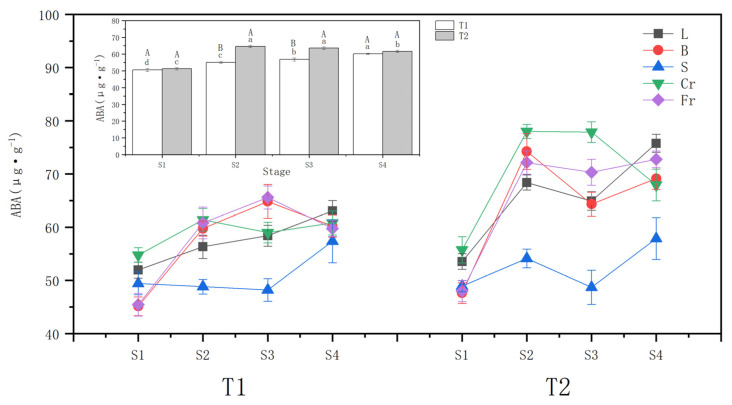
Effects of different drought treatments on abscisic acid (ABA) content in whole plant and organs of *P. orientalis*, L: Leaf; B: Branchlet; S: Stem; Cr: Coarse root; Fr: Fine root. T1: Rapid drought–rehydration group; T2: Slow drought–rehydration group. Different letters indicate significant differences in measurement indicators between the treatment group (*p* < 0.05), lowercase letters indicate differences in different stages of the same treatment, and uppercase letters indicate differences in different treatments at the same stage.

**Figure 4 biology-13-00194-f004:**
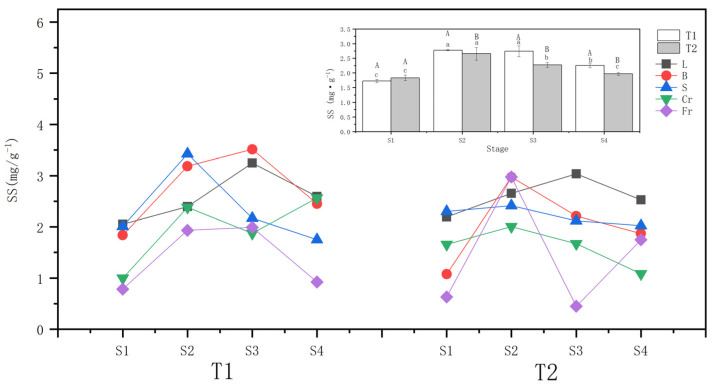
Effects of different drought treatments on the soluble sugar content of the whole plant and organs of *P. orientalis*. L: Leaf; B: Branchlet; S: Stem; Cr: Coarse root; Fr: Fine root. T1: Rapid drought–rehydration group; T2: Slow drought–rehydration group. Different letters indicate significant differences in measurement indicators between the treatment group (*p* < 0.05), lowercase letters indicate differences in different stages of the same treatment, and uppercase letters indicate differences in different treatments at the same stage. The explanations of T1 and T2 are shown in Figure 2.

**Figure 5 biology-13-00194-f005:**
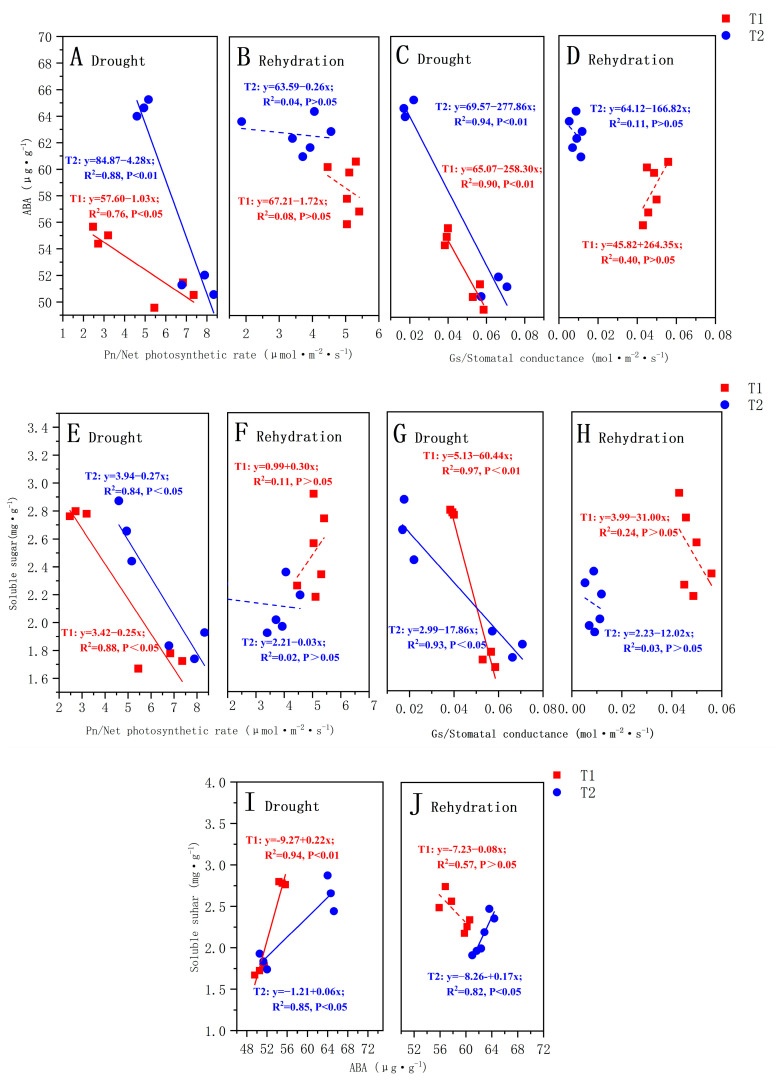
The relationship between the ABA content and net photosynthetic rate (P_n_) during drought (**A**) and rehydration (**B**); The relationship between the ABA content and stomatal conductance (G_s_) during drought (**C**) and rehydration (**D**); The relationship between the SS content and net photosynthetic rate (P_n_) during drought (**E**) and rehydration (**F**); The relationship between the SS content and stomatal conductance (G_s_) during drought (**G**) and rehydration (**H**); The relationship between the ABA content and SS content during drought (**I**) and rehydration (**J**). The solid lines identify significant correlations. The Pearson’s correlation coefficients (R^2^) and significance levels are shown. The explanations of T1 and T2 are shown in Figure 2.

**Table 1 biology-13-00194-t001:** Basic information in test materials of *Platycladus orientalis* (L.).

	Forest Age (a)	Average Base Diameter (mm)	Average Tree Height (cm)
*Platycladus orientalis*	3	12.14 ± 1.71	123.63 ± 8.65

**Table 2 biology-13-00194-t002:** Basic properties of soil.

Soil Bulk Density (g/cm^3^)	Soil Volumetric Moisture Content (%)	Total Nitrogen (g/kg)	Total Phosphorus (g/kg)	Total Potassium (g/kg)
1.44	28	1.137	1.059	18.21

## Data Availability

The data presented in this study are available in the article.

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
