# Peer review of "The Response of Endogenous ABA and Soluble Sugars of Platycladus orientalis to Drought and Post-Drought Rehydration"

_biology, 2024, doi:10.3390/biology13030194_

Round 1

Reviewer 1 Report

Comments and Suggestions for Authors

Dear Authors

The work is interesting and well conducted, however, some changes are necessary.

Line 18 “abridged” please rephrase

Line 15 “Organ” specify please

Line 15 “drought-rehydration (T1) and slow drought-rehydration (T2) treatments”  specify please

Line 36 “varying” ? which one?

Line 52 Add the following sentence; “The repercussions of drought stress on plants can be evaluated by monitoring physiological or biochemical indicators (Cataldo et al., 2023).

Cataldo, E., Fucile, M., Manzi, D., Peruzzi, E., & Mattii, G. B. (2023). Effects of Zeowine and compost on leaf functionality and berry composition in Sangiovese grapevines. The Journal of Agricultural Science, 1-16.

Line 55 Add the following sentence; “However, when a plant is subjected to drought conditions, the net photosynthesis, the water potential, and the stomatal conductance are altered, making photosynthesis a crucial indicator of plant stress resilience (Cataldo et al., 2023b)”

Cataldo, E., Fucile, M., Manzi, D., Masini, C. M., Doni, S., & Mattii, G. B. (2023b). Sustainable soil management: effects of clinoptilolite and organic compost soil application on eco-physiology, quercitin, and hydroxylated, methoxylated anthocyanins on Vitis vinifera. Plants, 12(4), 708.

Line 138 the authors could provide more information on the pots (type of soil, monitoring conditions, etc

Lines 147-163 authors could provide more information and more details

for each parameter measured or sampled the authors should provide the number of repetitions and the sampling methods

Comments on the Quality of English Language

Moderate editing of English language required

Reviewer 2 Report

Comments and Suggestions for Authors

This serves as my review report for the manuscript by Zhao et al., on "The responses of endogenous ABA and soluble sugars of Platycladus orientalis to drought and post-drought rehydration".

ABSTRACT

please revise the abstract to improve on quality.

INTRODUCTION

I found the introduction well-written and appropriate for the research topic and study. Below are a few comments

1.  At the end of the first paragraph of the introduction, following the statement in Lines 45-57, the authors need to explain why it is important to clarify the physiological mechanisms of drought in trees. The question is what's next after clarifying these mechanisms.

2. The word "reflect" in Line 48 needs revision. 

3. In Line 63, I suggest that  the term "stress-resistant genes" is replaced with "stress-responsive genes" because not all genes that are expressed in response to drought are stress-resistant.

4. I do not quite understand the meaning of Line 66-68 relating to xylem embolism.

5. Maybe after Lines 71-72, the authors can briefly define non-structural carboxylation and its end products.

6. Please delete the sentence in Lines 72-73  "The drought process may also affect NCS dynamics".

7. please provide an explanation of the rationale behind the differential storage of sugars in roots and leaves during different drought intensities as stated in Lines 80- 82. 

8. In Lines 84 -86, please substantiate your facts by adding reference citations. The same with Lines 93-94. In Lines 93, I would delete the words "domestic and foreign". In the same sentence, I would also revise the word "only" and replace it with a more cautious word like "mostly" and add refs at the end of that sentence.

9. The second half of the sentence in Lines 89-92 needs revision. The sentence in Lines 97-100 also needs revision for improved clarity.

10. The sentence in Lines 106-107 also needs revision and it should really read as the aim of the study. In the sam sentence, the term "experimental materials" required revision. In fact, the entire last paragraph of the introduction in Lines 106-115 requires extensive revision. Water dynamics needs to be reworded for improved clarity. The scientific name in Lines 112 and throughout the manuscript should be written in italics.

MATERIALS & METHODS

1. Please add a ref at the end of the sentence in Lines 120-123.

2. Please check if the term "field-cultivated soil" is technically correct.

3. The term "previous year" requires substantiating. Previous to what happening?

4. In line 132 - 133, how many plant species did you experiment on in this study?

5. In Table 1, you need to improve on the column headings. I also suggest that the units of measurements be written in brackets because the forward slash  (\ ) is complicating the column headings.

6. Just a general comment, mention of months in such studies only makes sense to readers who are familiar with your geographical area. in my view, it would be better to mention the month and the main characteristic features of the period (line 133).

7. "Long period" needs to be more specific (Line 137).

8. In Line 144-145, please revise sentence construction because you "cannot stop trees from irrigation". 

9. At the end of the sentences in lines 148 -149 & 156 - 157, please add the words " during the rehydration phase".

10. How is the time read in Figure 1? e.g. what does 8/3 mean? and what are the units of time in this figure?

11. Either a reference for the ABA assay should be give or a more detailed protocol written (Line 184-192). 

12. In Line 194, please specify the samples. Also cite a ref for the soluble sugar content assay.

13. In line 205, the word "organised" needs revision. Did you want to say "analysed".

RESULTS

1. In line 217, please add the word "the" before "different".

2. In lines 224 and 225, there is a word missing after stomata (before the comma in Line 224), and at the end of Line 225. 

3. In the results section, the P values should be written with a small letter p and a sign for "less than or equal to" OR "greater than or equal to" depending on the context.

4. In Line 228, please replace "and" with "upon". In Lines 226-229, I do not understand how FC declines with rehydration. please recheck the sentence.

5. Please formulate a composite legend for Figure 2. Also check on eligibility of the text in this figure.

6. I think the words "same below" in line 238 are misplaced. please check.

7. Please recheck if the S2 mentioned in line 270 is correct. I would also delete the word moment in the manuscript, and leave the text as "at S1, S2, S3 or S4".

8. In Figure 5 A-D, what are the parameters and their units of measurement of the x-axes? This is complicating my understanding for the Figure and corresponding text in parts of section 3.4

DISCUSSION

1. In Line 331, what threshold in this?

2. At the end of Line 337, please add the word "deficits" after water.

3. In Line 340, please add the word "possibly" after T2.

4. Please check if the text in Line 339-342 does not conflict that in Lines 343-345.

5. The word "dissipation' is line 363 requires revision. I would use "loss"

6. The word "resistance" in Line 369 also  requires revision. I would use "response".

7. since the study did not measure cell membrane damage at all (unless if I missed it), I would delete the mention of of mitigation of cell membrane damage in Line 370.

8. In Line 399, delete "regular".

9. Please cite refs at the end of the sentence in Lines 409-411.

10. I do not understand the meaning of the word "embolism".

11. add refs of the studies in question at the end of Lines 459-461.

12. In Line 461-463, why would plants communicate drought information to the soil? please check sentence construction and the preposition used.

13. I do not know how the conclusion in lines 468 - 469 was reached.

CONCLUSION

1. In line 502, please replace the word "object" with "plant".

2. I am not sure how the authors came to the conclusion in Lines  514 - 517.

Comments on the Quality of English Language

The quality of English is generally good. Only minor edits are required. However, the abstract needs to be re-written.

Reviewer 3 Report

Comments and Suggestions for Authors

First of all, I would like to congratulate the authors for such wonderful work. I have some minor comments that will add more clarity to the manuscript, as follows: 

1) Add more reference citations in the Introduction section. For example, you mentioned in lines 86 to 105 about many experimental studies, but no references for those studies are provided in the manuscript. 

2) In the Results section, add exact percentages to indicate the amount of content that increased or decreased. For instance, in line 251. 

3) Improve your discussion by adding references and comparing more with previous studies.

4) Needs to improve the English language in the entire manuscript.

Comments on the Quality of English Language

It needs improvement.

Round 2

Reviewer 1 Report

Comments and Suggestions for Authors

Accept

Comments on the Quality of English Language

Minor editing of English language required